# VLMO: Unified Vision-Language Pre-Training with Mixture-of-Modality-Experts

**Hangbo Bao**[1]*, **Wenhui Wang**[2], **Li Dong**[2], **Qiang Liu**[2], **Owais Khan Mohammed**[2],
**Kriti Aggarwal**[2], **Subhojit Som**[2], **Songhao Piao**[1], **Furu Wei**[2]
[1]Harbin Institute of Technology, [2]Microsoft Corporation
`https://aka.ms/msragi`

## Abstract

We present a unified **V**ision-**L**anguage pretrained **Mo**del (**VLMO**) that jointly learns a dual encoder and a fusion encoder with a modular Transformer network. Specifically, we introduce Multiway Transformer, where each block contains a pool of modality-specific experts and a shared self-attention layer. Because of the modeling flexibility of Multiway Transformer, pretrained VLMO can be fine-tuned as a fusion encoder for vision-language classification tasks, or used as a dual encoder for efficient image-text retrieval. Moreover, we propose a stagewise pre-training strategy, which effectively leverages large-scale image-only and text-only data besides image-text pairs. Experimental results show that VLMO achieves state-of-the-art results on various vision-language tasks, including VQA, NLVR2 and image-text retrieval. The code and pretrained models are available at `http://aka.ms/vlmo`.

## 1 Introduction

Vision-Language (VL) pre-training [31, 42, 36, 27, 21, 24] learns generic cross-modal representations from large-scale image-text pairs. Previous models usually employ image-text matching, image-text contrastive learning, masked region classification/feature regression, word-region/patch alignment and masked language modeling to aggregate and align visual and linguistic information. Then the pretrained models can be directly fine-tuned on downstream vision-language tasks, such as VL retrieval and classification (visual question answering, visual reasoning, etc.).

Two mainstream architectures are widely used in previous work. CLIP [36] and ALIGN [19] adopt a *dual-encoder* architecture to encode images and text separately. Modality interaction is handled by the cosine similarity of the image and text feature vectors. The dual-encoder architecture is effective for retrieval tasks, especially for masses of images and text. Feature vectors of images and text can be pre-computed and stored. However, the shallow interaction between images and text is not enough to handle complex VL classification tasks. ViLT [21] finds that CLIP gives a relatively low accuracy on visual reasoning task. Another line of work [31, 42, 44, 4, 21, 24] relies on a fusion encoder with cross-modal attention to model image-text pairs. Multi-layer Transformer [46] networks are usually employed to fuse image and text representations. The *fusion-encoder* architecture achieves superior performance on VL classification tasks. But it requires to jointly encode all possible image-text pairs to compute similarity scores for retrieval tasks. The quadratic time complexity leads to a much slower inference speed than the dual-encoder models whose time complexity is linear.

In order to take advantage of the two types of architectures, we propose a unified **V**ision-**L**anguage pretrained **Mo**del (**VLMO**) that can be used as either a dual encoder to separately encode images and text for retrieval tasks, or used as a fusion encoder to model the deep interaction of image-text pairs for

---

*Contribution during internship at Microsoft.

36th Conference on Neural Information Processing Systems (NeurIPS 2022).

classification tasks. This is achieved by introducing Multiway Transformer that can encode various modalities (images, text, and image-text pairs) within a Transformer block. Multiway Transformer employs a pool of modality experts to replace the feed-forward network in standard Transformer. It captures modality-specific information by switching to different modality experts, and uses the shared self-attention across modalities to align visual and linguistic information. Specifically, Multiway Transformer consists of three modality experts, namely vision expert for image encoding, language expert for text encoding, and vision-language expert for image-text fusion. Thanks to the modeling flexibility, we can reuse Multiway Transformer with the shared parameters for different purposes, i.e., text-only encoder, image-only encoder, and image-text fusion encoder.

VLMO is jointly learned with three pre-training tasks, namely image-text contrastive learning, image-text matching, and masked language modeling. In addition, we propose a stagewise pre-training strategy to effectively leverage large-scale image-only and text-only corpus besides image-text pairs in VLMO pre-training. We first pretrain vision experts and self-attention modules of Multiway Transformer on image-only data using masked image modeling proposed in BEIT [3]. We then pretrain language experts on text-only data using masked language modeling [11]. Finally, the model is used to initialize vision-language pre-training. By getting rid of the limited size of image-text pairs and their simple and short captions, stagewise pre-training on large amounts of image-only and text-only data helps VLMO to learn more generalizable representations.

Experimental results demonstrate that VLMO achieves state-of-the-art results on vision-language retrieval and classification tasks. Our model, used as a dual encoder, outperforms fusion-encoder-based models [4, 15, 21, 24] while enjoying a much faster inference speed on retrieval tasks. Moreover, our model also achieves state-of-the-art results on visual question answering (VQA) and natural language for visual reasoning (NLVR2), where VLMO is used as a fusion encoder.

Our main contributions are summarized as follows:

- We propose a unified vision-language pretrained model VLMO that can be used as a fusion encoder for classification tasks, or fine-tuned as a dual encoder for retrieval tasks.

- We introduce a general-purpose multimodal Transformer for vision-language tasks, namely Multiway Transformer, to encode different modalities. It captures modality-specific information by modality experts, and aligns contents of different modalities by the self-attention module shared across modalities.

- We show that stagewise pre-training using large amounts of image-only and text-only data greatly improves our vision-language pretrained model.

## 2   Related Work

Pre-training with Transformer [46] backbone networks has substantially advanced the state of the art across natural language processing [35, 11, 29, 23, 12, 37, 2, 8, 9, 5–7, 32], computer vision [13, 45, 3] and vision-language [44, 42, 4, 51, 36, 19, 21, 24, 47] tasks.

The approaches of vision-language pre-training can be divided into two categories. The first category utilizes a dual encoder to encode images and text separately, and uses cosine similarity or a linear projection layer to model the interaction between images and text [36, 19]. Image-text contrastive learning is usually employed to optimize the model. Dual-encoder models are effective for vision-language retrieval tasks. However, the simple interaction is not enough to handle tasks that require complex reasoning, such as visual reasoning and visual question answering (VL classification tasks). The second category models the interaction of images and text using a deep fusion encoder with cross-modal attention [44, 31, 42, 25, 53, 4, 27, 26, 15, 51, 17, 18, 21, 24, 48]. Image-text matching, masked language modeling, word-region/patch alignment, masked region classification and feature regression are widely used to train fusion-encoder-based models. These models achieve better performance for vision-language classification tasks, while the joint encoding of all image-text pairs leads to a slow inference speed for retrieval tasks. A large portion of fusion-encoder-based models rely on an off-the-shelf object detector like Faster R-CNN [38] to obtain image region features. Generating region features slows down the inference speed and renders the approach less scalable. Recently, Pixel-BERT [17] removes object detector and encodes images into grid features by convolutional neural networks. ALBEF [24] employs image Transformer [13, 45] to obtain the representations of images, and uses text Transformer [11] to learn the contextualized representations

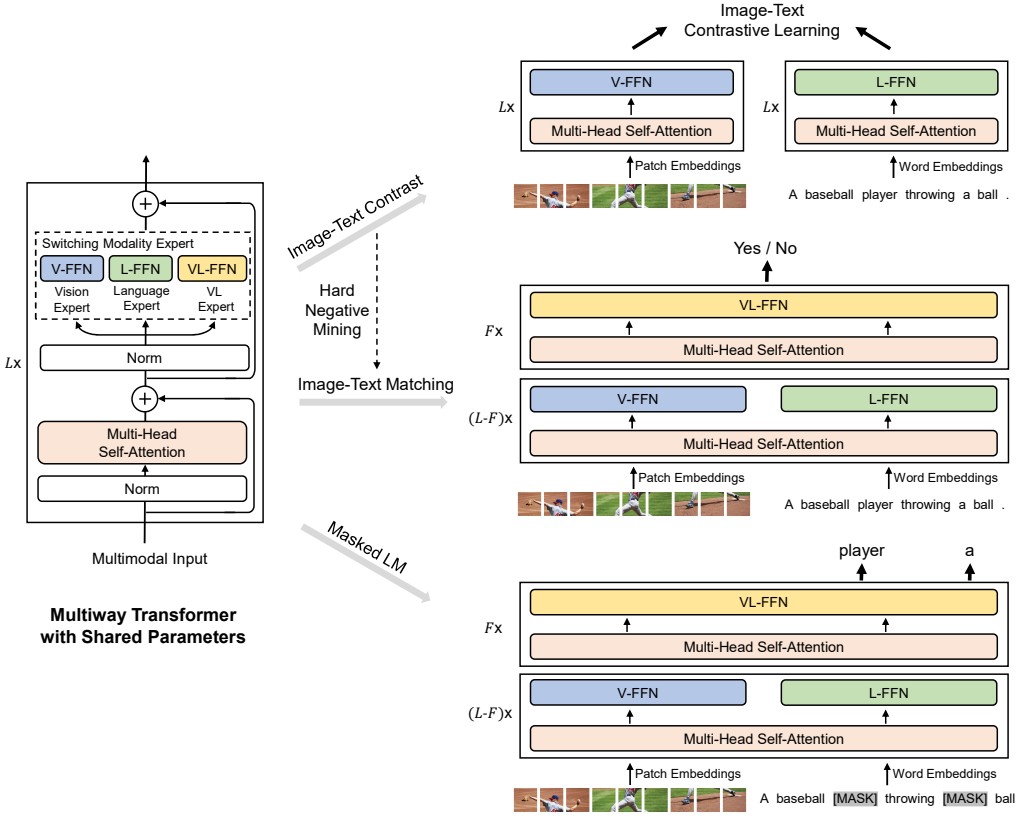

Figure 1: Overview of VLMo pre-training. We introduce Multiway Transformer to encode different modality input by modality-specific experts. The model parameters are shared across image-text contrastive learning, masked language modeling, and image-text matching pre-training tasks. During fine-tuning, the flexible modeling enables us to use VLMo as either a dual encoder (i.e., separately encode images and text for retrieval tasks) or a fusion encoder (i.e., jointly encode image-text pairs for better interaction across modalities).

of text. These representations are then fused by cross-modal attention. ViLT [21] encodes images into patch embeddings, and then feed the concatenation of image patch embeddings and word embeddings into a Transformer network to learn contextualized representations and model the interaction of images and text.

Different from previous work, our unified pre-training using shared Multiway Transformer enables the model perform separate encoding for retrieval tasks, and jointly encode image-text pairs to capture deeper interaction for classification tasks. Our model achieves competitive performance, while enjoying a faster inference speed for both retrieval and classification tasks.

## 3 Methods

Given image-text pairs, VLMo obtains image-only, text-only and image-text pair representations by the Multiway Transformer network. As shown in Figure 1, the unified pre-training optimizes shared Multiway Transformer with image-text contrastive learning on image-only and text-only representations, image-text matching and masked language modeling on image-text pair representations. Thanks to the modeling flexibility, the model can be used as a dual encoder for retrieval tasks to encode images and text separately during fine-tuning. It can also be fine-tuned as a fusion encoder to model deeper modality interaction of images and text for classification tasks.

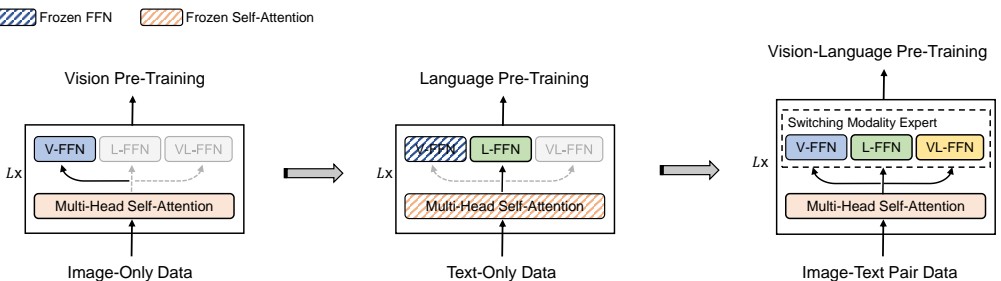

Figure 2: Stagewise pre-training using image-only and text-only corpora. We first pretrain the vision expert (V-FFN) and self-attention module on large-scale image-only data as in BEiT [3]. Then the parameters of vision expert and self-attention module are frozen, and we train the language expert (L-FFN) by masked language modeling on large amounts of text-only data. Finally, we train the whole model with vision-language pre-training.

## 3.1 Input Representations

Given an image-text pair, we encode the pair into image, text and image-text vector representations. These representations are then fed into the Multiway Transformer to learn contextualized representations and align image and text feature vectors.

**Image Representations**    Following vision Transformers [13, 45, 3], the 2D image $v \in \mathbb{R}^{H \times W \times C}$ is split and reshaped into $N = HW/P^2$ patches $v^p \in \mathbb{R}^{N \times (P^2 C)}$, where $C$ is the number of channels, $(H, W)$ is the resolution of the input image, and $(P, P)$ is the patch resolution. The image patches are then flattened into vectors and are linearly projected to obtain patch embeddings. We also prepend a learnable special token [I_CLS] to the sequence. Finally, image input representations are obtained via summing patch embeddings, learnable 1D position embeddings $V_{pos} \in \mathbb{R}^{(N+1) \times D}$ and image type embedding $V_{type} \in \mathbb{R}^D$: $H_0^v = [v_{[\text{I\_CLS}]}, V v_i^p, \ldots, V v_N^p] + V_{pos} + V_{type}$, where $H_0^v \in \mathbb{R}^{(N+1) \times D}$, linear projection $V \in \mathbb{R}^{(P^2 C) \times D}$.

**Text Representations**    Following BERT [11], we tokenize the text to subword units by Word-Piece [49]. A start-of-sequence token ([T_CLS]) and a special boundary token ([T_SEP]) are added to the text sequence. Text input representations $H_0^w \in \mathbb{R}^{(M+2) \times D}$ are computed via summing the corresponding word embedding, text position embedding and text type embedding $H_0^w = [w_{[\text{T\_CLS}]}, w_i, \ldots, w_M, w_{[\text{T\_SEP}]}] + T_{pos} + T_{type}$. $M$ indicates the length of tokenized subword units.

**Image-Text Representations**    We concatenate image and text input vectors to form the image-text input representations $H_0^{vl} = [H_0^w; H_0^v]$

## 3.2 Multiway Transformer

Inspired by mixture-of-experts networks [41, 14], we propose a general-purpose multimodal Transformer for vision-language tasks, namely Multiway Transformer, to encode different modalities. Multiway Transformer introduces mixture of modality experts as a substitute of the feed forward network of standard Transformer. Each modality expert is also the feed forward network which consists of two linear transformations and an activation. Given previous layer's output vectors $H_{l-1}, l \in [1, L]$, each Multiway Transformer block captures modality-specific information by switching to different modality expert, and employs multi-head self-attention (MSA) shared across modalities to align visual and linguistic contents. LN is short for layer normalization.

$$H_l' = \text{MSA}(\text{LN}(H_{l-1})) + H_{l-1} \tag{1}$$

$$H_l = \text{Multiway-FFN}(\text{LN}(H_l')) + H_l' \tag{2}$$

Multiway-FFN selects an expert among multiple modality experts to process the input according to the modality of the input vectors $H_l'$ and the index of the Transformer layer. Specifically, there

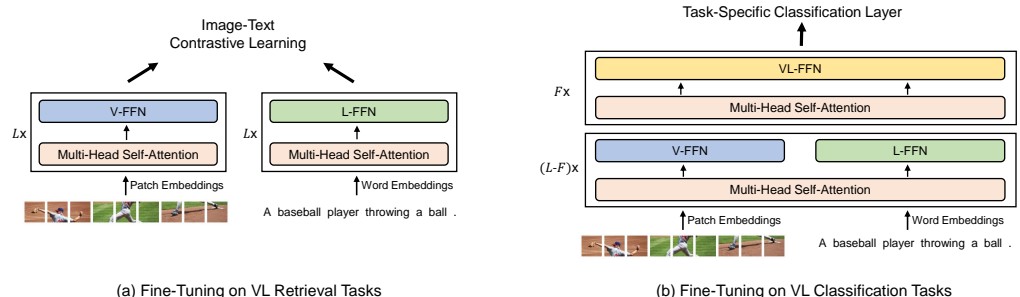

(a) Fine-Tuning on VL Retrieval Tasks  (b) Fine-Tuning on VL Classification Tasks

Figure 3: Fine-tuning VLMo on vision-language retrieval and classification tasks. The model can be fine-tuned as a dual encoder to separately encode image and text for retrieval tasks. VLMo can also be used as a fusion encoder to handle interaction of image-text pairs for classification tasks.

are three modality experts: vision expert (V-FFN), language expert (L-FFN) and vision-language expert (VL-FFN). If the input is image-only or text-only vectors, we use vision expert for encoding images and language expert for encoding text. If the input consists of vectors of multiple modalities, such as the vectors of image-text pair, we employ vision expert and language expert to encode the respective modality vectors at the bottom Transformer layers. Vision-language expert is then used at the top layers to capture more modality interaction. Compared with conventional mixture-of-experts networks [41, 14], Multiway Transformer conducts hard routing according the input modality. Given the three types of input vectors, we obtain image-only, text-only and image-text contextualized representations.

## 3.3   Pre-Training Tasks

VLMo is jointly pretrained by image-text contrastive learning on the image and text representations, masked language modeling and image-text matching on the image-text pair representations with shared parameters.

**Image-Text Contrast**   Given a batch of $N$ image-text pairs, image-text contrastive learning aims to predict the matched pairs from $N \times N$ possible image-text pairs. There are $N^2 - N$ negative image-text pairs within a training batch.

The final output vectors of `[I_CLS]` token and `[T_CLS]` token are used as the aggregated representation of the image and text, respectively. Followed by a linear projection and normalization, we obtain image vectors $\{\hat{\boldsymbol{h}}_i^v\}_{i=1}^N$ and text vectors $\{\hat{\boldsymbol{h}}_i^w\}_{i=1}^N$ in a training batch to compute image-to-text and text-to-image similarities:

$$s_{i,j}^{i2t} = \hat{\boldsymbol{h}}_i^{v\mathsf{T}}\hat{\boldsymbol{h}}_j^w, \ s_{i,j}^{t2i} = \hat{\boldsymbol{h}}_i^{w\mathsf{T}}\hat{\boldsymbol{h}}_j^v \tag{3}$$

$$p_i^{i2t} = \frac{\exp(s_{i,i}^{i2t}/\sigma)}{\sum_{j=1}^N \exp(s_{i,j}^{i2t}/\sigma)}, \ p_i^{t2i} = \frac{\exp(s_{i,i}^{t2i}/\sigma)}{\sum_{j=1}^N \exp(s_{i,j}^{t2i}/\sigma)} \tag{4}$$

Where $s_{i,j}^{i2t}$ represents image-to-text similarity of image of $i$-th pair and text of $j$-th pair, $s_{i,j}^{t2i}$ is the text-to-image similarity. $\hat{\boldsymbol{h}}_i^w \in \mathbb{R}^D$ and $\hat{\boldsymbol{h}}_j^v \in \mathbb{R}^D$ indicate the normalized vectors of $i$-th text and $j$-th image, $\sigma$ is a learned temperature parameter. $p_i^{i2t}$ and $p_i^{t2i}$ are the softmax-normalized similarities. Cross-entropy losses over image-to-text and text-to-image similarities are used to train the model.

**Masked Language Modeling**   Following BERT [11], we randomly choose tokens in the text sequence, and replace them with the `[MASK]` token. The model is trained to predict these masked tokens from all the other unmasked tokens and vision clues. We use 15% masking probability as in BERT. The final output vectors of masked tokens are fed into a classifier over the whole text vocabulary with cross-entropy loss.

**Image-Text Matching**   Image-text matching aims to predict whether the image and text is matched. We use the final hidden vector of the `[T_CLS]` token to represent the image-text pair, and feed the

vector into a classifier with cross-entropy loss for binary classification. Inspired by ALBEF [24], we sample hard negative image-text pairs based on the contrastive image-to-text and text-to-image similarities.

### 3.4 Stagewise Pre-Training

We introduce a stagewise pre-training strategy, which leverages large-scale image-only and text-only corpus to improve the vision-language model. As present in Figure 2, we first perform vision pre-training on image-only data, and then perform language pre-training on text-only data to learn general image and text representations. The model is used to initialize the vision-language pre-training to learn the alignment of visual and linguistic information. For vision pre-training, we train the attention module and vision expert of Multiway Transformer as in BEiT [3] on image-only data. We directly utilize the pretrained parameters of BEiT to initialize the attention module and vision expert. For language pre-training, we freeze parameters of the attention module and vision expert to avoid catastrophic forgetting of vision knowledge learned in the first stage, and utilize masked language modeling [11] to optimize the language expert on text-only data. Compared with image-text pairs, image-only and text-only data are easier to collect. In addition, text data of image-text pairs is usually short and simple. Pre-training on image-only and text-only corpus improves the generalization on complex pairs.

### 3.5 Fine-Tuning VLMo on Downstream Tasks

As present in Figure 3, our model can be fine-tuned to adapt to various vision-language retrieval and classification tasks.

**Vision-Language Classification**  For classification tasks such as visual question answering and visual reasoning, VLMo is used as a fusion encoder to model modality interaction of images and text. We use the final encoding vector of the token `[T_CLS]` as the representation of the image-text pair, and feed it to a task-specific classifier layer to predict the label.

**Vision-Language Retrieval**  For retrieval tasks, VLMo can be used as a dual encoder to encode images and text separately. During fine-tuning, our model is optimized for the image-text contrastive loss. During inference, we compute representations of all images and text, and then use dot product to obtain image-to-text and text-to-image similarity scores of all possible image-text pairs. Separate encoding enables a much faster inference speed than fusion-encoder-based models.

## 4 Experiments

We pretrain our model using large-scale image-text pairs and evaluate the model on visual-linguistic classification and retrieval tasks.

### 4.1 Pre-Training Setup

Following previous work [4, 21], our pre-training data consists of four image captioning datasets: Conceptual Captions (CC) [40], SBU Captions [33], COCO [28] and Visual Genome (VG) [22] datasets. There are about $4M$ images and $10M$ image-text pairs in the pre-training data.

Our models adopt the same network configuration as ViT [13] and BEiT [3]. VLMo-Base consists of 12-layer Transformer blocks with $768$ hidden size and $12$ attention heads. VLMo-Large is a 24-layer Transformer network with $1024$ hidden size and $16$ attention heads. VLMo-Base uses vision-language expert on the top two Transformer layers, and VLMo-Large introduces vision-language expert on the top three layers. VLMo-Base consists of 175M parameters and VLMo-Large contains 562M parameters. For images, the input resolution is $224 \times 224$ and the patch size is $16 \times 16$ during pre-training. We apply RandAugment [10] to the input images. The tokenizer of the uncased version of BERT is employed to tokenize the text. The maximum text sequence length is set to $40$. We also employ whole word masking for the masked language modeling pre-training task. We pretrain the models for 200k steps with 1024 batch size. We utilize AdamW [30] optimizer with $\beta_1 = 0.9$, $\beta_2 = 0.98$. The peak learning is 2e-4 for the base-size model, 5e-5 for the large-size model. Weight decay is set to $0.01$. We use linear warmup over the first 2.5k steps and linear decay.

| Model | # Pretrain Images | VQA | | NLVR2 | |
|---|---|---|---|---|---|
| | | test-dev | test-std | dev | test-P |
| *Base-Size Models Pretrained on COCO, VG, SBU and CC datasets* | | | | | |
| UNITER-Base [4] | 4M | 72.70 | 72.91 | 77.18 | 77.85 |
| VILLA-Base [15] | 4M | 73.59 | 73.67 | 78.39 | 79.30 |
| UNIMO-Base [26] | 4M | 73.79 | 74.02 | - | - |
| ViLT-Base [21] | 4M | 71.26 | - | 75.70 | 76.13 |
| ALBEF-Base [24] | 4M | 74.54 | 74.70 | 80.24 | 80.50 |
| **VLMO-Base** | 4M | **76.64** | **76.89** | **82.77** | **83.34** |
| *Large-Size Models Pretrained on COCO, VG, SBU and CC datasets* | | | | | |
| UNITER-Large [4] | 4M | 73.82 | 74.02 | 79.12 | 79.98 |
| VILLA-Large [15] | 4M | 74.69 | 74.87 | 79.76 | 81.47 |
| UNIMO-Large [26] | 4M | 75.06 | 75.27 | - | - |
| **VLMO-Large** | 4M | **79.94** | **79.98** | **85.64** | **86.86** |
| *Models Pretrained on More Data* | | | | | |
| VinVL-Large [51] | 5.7M | 76.52 | 76.60 | 82.67 | 83.98 |
| SimVLM-Large [48] | 1.8B | 79.32 | 79.56 | 84.13 | 84.84 |
| SimVLM-Huge [48] | 1.8B | 80.03 | 80.34 | 84.53 | 85.15 |
| Florence-Huge [50] | 900M | 80.16 | 80.36 | - | - |
| Flamingo [1] | 2.3B | 82.00 | 82.10 | - | |
| **VLMO-Large++** | 1.0B | **82.88** | **82.78** | **88.62** | **89.54** |

Table 1: Fine-tuning results of base-size and large-size VLMO on vision-language classification datasets. VLMO-Large++ is the model trained on one billion noisy image-text pairs with a larger batch size. We report vqa-score on VQA test-dev and test-standard split, and report accuracy for NLVR2 development and public test set (test-P).

## 4.2 Training on Larger-scale Datasets

We scale up vision-language representation learning by training VLMO-Large on one billion noisy web image-text pairs with a larger batch size. We first pretrain the model for 200k steps with 16k batch size, and then continue train the model for 100k steps with 32k batch size. The other hyper-parameters are the same as the training on 4M data. Please refer to the supplementary material for more details of hyper-parameters used for pre-training and fine-tuning.

## 4.3 Evaluation on Vision-Language Classification Tasks

We first conduct fine-tuning experiments on two widely used classification datasets: visual question answering [16] and natural language for visual reasoning [43]. The model is fine-tuned as a fusion encoder to model deeper interaction.

**Visual Question Answering (VQA)**    For VQA, a natural image and a question are given, the task is to generate/choose the correct answer. We train and evaluate the model on VQA 2.0 dataset [16]. Following common practices, we convert VQA 2.0 to a classification task, and choose the answer from a shared set consists of $3,129$ answers. We use the final encoding vector of the [T_CLS] token as the representation of the image-question pair and feed it to a classifier layer to predict the answer.

**Natural Language for Visual Reasoning (NLVR2)**    The NLVR2 [43] dataset requires the model to predict whether a text description is true about a pair of images. Following OSCAR [27] and VinVL [51], we convert the triplet input to two image-text pairs, each containing the text description and one image. We concatenate the final output vectors of the [T_CLS] token of the two input pairs. The concatenated vector is then fed into a classification layer to predict the label.

We present the results of VL classification tasks in Table 1. VLMO achieves state-of-the-art performance and substantially outperforms previous methods. Our large-size model even outperforms SimVLM-Huge [48] and Florence-Huge [50] by a large margin, which consists of more parameters and are also trained on larger-scale image-text pairs. Our model uses a simple linear projection to embed images as in ViLT [21]. This leads to a significant speedup compared with previous models using image region features, which are extracted by an off-the-shelf object detector [31, 42, 4, 15, 26, 51].

| Model | # Pretrain Images | MSCOCO (5K test set) | | | | | | Flickr30K (1K test set) | | | | | |
|---|---|---|---|---|---|---|---|---|---|---|---|---|---|
| | | TR | | | IR | | | TR | | | IR | | |
| | | R@1 | R@5 | R@10 | R@1 | R@5 | R@10 | R@1 | R@5 | R@10 | R@1 | R@5 | R@10 |
| *Base-Size Models Pretrained on COCO, VG, SBU and CC datasets* | | | | | | | | | | | | | |
| UNITER-Base | 4M | 64.4 | 87.4 | 93.1 | 50.3 | 78.5 | 87.2 | 85.9 | 97.1 | 98.8 | 72.5 | 92.4 | 96.1 |
| VILLA-Base | 4M | - | - | - | - | - | - | 86.6 | 97.9 | 99.2 | 74.7 | 92.9 | 95.8 |
| ViLT-Base | 4M | 61.5 | 86.3 | 92.7 | 42.7 | 72.9 | 83.1 | 83.5 | 96.7 | 98.6 | 64.4 | 88.7 | 93.8 |
| ALBEF-Base‡ | 4M | 73.1 | 91.4 | 96.0 | 56.8 | 81.5 | 89.2 | **94.3** | **99.4** | 99.8 | **82.8** | **96.7** | **98.4** |
| **VLMo-Base**† | 4M | **74.8** | **93.1** | **96.9** | **57.2** | **82.6** | **89.8** | 92.3 | **99.4** | **99.9** | 79.3 | 95.7 | 97.8 |
| *Large-Size Models Pretrained on COCO, VG, SBU and CC datasets* | | | | | | | | | | | | | |
| UNITER-Large | 4M | 65.7 | 88.6 | 93.8 | 52.9 | 79.9 | 88.0 | 87.3 | 98.0 | 99.2 | 75.6 | 94.1 | 96.8 |
| VILLA-Large | 4M | - | - | - | - | - | - | 87.9 | 97.5 | 98.8 | 76.3 | 94.2 | 96.8 |
| **VLMo-Large**† | 4M | **78.2** | **94.4** | **97.4** | **60.6** | **84.4** | **91.0** | **95.3** | **99.9** | **100.0** | **84.5** | **97.3** | **98.6** |
| *Models Pretrained on More Data* | | | | | | | | | | | | | |
| VinVL-Large | 5.7M | 75.4 | 92.9 | 96.2 | 58.8 | 83.5 | 90.3 | - | - | - | - | - | - |
| ALIGN-Large† | 1.8B | 77.0 | 93.5 | 96.9 | 59.9 | 83.3 | 89.8 | 95.3 | 99.8 | **100.0** | 84.9 | 97.4 | 98.6 |
| Florence-Huge† | 900M | 81.8 | 95.2 | - | 63.2 | 85.7 | - | 97.2 | 99.9 | - | 87.9 | 98.1 | - |
| **VLMo-Large++**† | 1.0B | **83.1** | **96.0** | **98.2** | **65.2** | **86.5** | **92.2** | 96.8 | **100.0** | **100.0** | **88.1** | **98.4** | **99.3** |

Table 2: Fine-tuning results of text-retrieval (TR) and image-retrieval (IR) on COCO and Flickr30K. †: ALIGN, Florence and our model encode images and text separately, and then employ a shallow interaction (dot product) to obtain the similarity scores. ‡: ALBEF first encodes images and text separately to obtain the top-$k$ candidates, and then feed these representations into a fusion encoder to rerank the candidates. The others require to encode all image-text combinations by a fusion encoder. VLMo-Large++ represents the model trained on one billion noisy image-text pairs with a larger batch size.

| Stagewise Pre-Training | NLVR2 | | Flickr30k | |
|---|---|---|---|---|
| | dev | test-P | TR | IR |
| Image-Only Pre-Training | 80.33 | 81.06 | 95.60 | 87.69 |
| Image-Only + Text-Only Pre-Training | **82.09** | **82.49** | **95.67** | **88.52** |

Table 3: Ablation studies of stagewise pre-training, i.e., different initialization for vision-language pre-training. We report the average of R@1, R@5 and R@10 for Flickr30k. Results of NLVR2 are averaged over three runs.

## 4.4 Evaluation on Vision-Language Retrieval Tasks

The retrieval tasks contain image-to-text retrieval and text-to-image retrieval. We evaluate the model on the widely used COCO [28] and Flickr30K [34] datasets, and use the Karpathy split [20] for both datasets. The model is used as a dual encoder for retrieval tasks. We encode images and text separately and compute their similarity scores by the dot product of image and text vectors.

As present in Table 2, VLMo achieves competitive performance with previous fusion-encoder-based models while having a much faster speed. Fusion-encoder-based models need to jointly encode all possible image-text pairs to compute their similarity scores, which requires quadratic time complexity. Moreover, our large-size model even outperforms the huge-size model of Florence [50], which also trained on massive image-text pairs using a larger batch size. VLMo pre-training can effectively leverage larger-scale noisy pairs and benefit from large batch training.

## 4.5 Evaluation on Vision Tasks

As shown in Table 4, we use VLMo as an image-only encoder and evaluate it on image classification (ImageNet [39]) and semantic segmentation (ADE20K [52]) tasks. The model also achieves competitive performance, even slightly better than the BEiT model used for the initialization of VLMo. The image resolution is 224×224 for ImageNet, and 512×512 for ADE20K. We perform intermediate fine-tuning [3] on ImageNet-21k for all three models.

| Models | ImageNet (acc@1) | ADE20K (mIoU) |
|---|---|---|
| VIT-Base | 83.6 | - |
| BEIT-Base | 85.2 | 52.8 |
| VLMO-Base | **85.5** | **53.4** |

Table 4: Results on image classification and semantic segmentation.

| Pre-Training Tasks | | | Transformer | | | NLVR2 | | Flickr30k | |
|---|---|---|---|---|---|---|---|---|---|
| ITC | ITM | MLM | Std TRM | Multiway | Multiway−VLExp | dev | test-P | TR | IR |
| ✓ | ✗ | ✗ | ✗ | ✓ | ✗ | 58.51 | 58.83 | 92.23 | 84.24 |
| ✓ | ✗ | ✓ | ✗ | ✓ | ✗ | 73.91 | 73.75 | 94.07 | 85.82 |
| ✓ | ✓ | ✗ | ✗ | ✓ | ✗ | 76.46 | 76.19 | 94.37 | 85.67 |
| ✓ | ✓ | ✓ | ✓ | ✗ | ✗ | 78.81 | 79.27 | 93.37 | 85.73 |
| ✓ | ✓ | ✓ | ✗ | ✗ | ✓ | 79.58 | 80.11 | 94.50 | 86.69 |
| ✓ | ✓ | ✓ | ✗ | ✓ | ✗ | **80.13** | **80.31** | **95.17** | **87.25** |

Table 5: Ablation studies of Multiway Transformer and vision-language pre-training tasks. "ITC" is short for image-text contrastive loss, "ITM" is image-text matching, and "MLM" is masked language modeling. "Std TRM" is short for standard Transformer, and "Multiway−VLExp" is Multiway Transformer without VL experts. The average of R@1, R@5 and R@10 is reported for Flickr30k. Results of NLVR2 are averaged over three runs.

## 4.6 Ablation Studies

**Stagewise Pre-Training** We first conduct ablation experiments of stagewise pre-training. ViLT [21] shows that using the ViT [13] model pretrained on image-only data as the initialization achieves better performance than the BERT model pretrained on text-only data. Therefore we start experiments with image-only pre-training. We compare using image-only pre-training, and image-only pre-training plus text-only pre-training as the initialization. For image-only pre-training, we directly use the parameters of BEIT-Base to initialize the self-attention module and all modality experts. For image-only pre-training plus text-only pre-training, we use pretrained parameters of BEIT-Base to initialize the vision expert and self-attention module of Multiway Transformer, and then pretrain its language expert on text corpora. As shown in Table 3, image-only pre-training plus text-only pre-training improves our vision-language model. We also have tried to perform vision-language pre-training with random initialization but obtain a relatively low accuracy on downstream tasks. Stagewise pre-training effectively leverages large-scale image-only and text-only corpus, and improves our vision-language pre-training. Moreover, given the limited size of image-text pairs we used during pre-training, stage-wise pre-training on image-only and text-only data alleviates the need for image-text pair data. We have tried to perform multitask training on image-only and text-only data to combine the first two stages and observe similar performance. Stage-wise pre-training effectively leverages the pretrained weights to reduce the computation cost.

**Multiway Transformer** We also conduct ablation experiments of Multiway Transformer. We employ ViT-Base to initialize the models for the ablation experiments. As present in Table 5, using Multiway Transformer achieves better performance than standard Transformer for both retrieval and classification tasks. In addition, we also analyse the contribution of vision-language expert (VL-FFN) used in Multiway Transformer. We remove the vision-language expert used in the top Transformer layers. Experimental results demonstrate that the introduction of vision-language expert improves the model. Using vision-language expert captures more modality interaction.

**Pre-Training Tasks** We perform ablation studies to analyse the contribution of different pre-training tasks, and the results are presented in Table 5. Compared with the model trained only using image-text contrastive loss, our unified training performs much better across classification and retrieval tasks. Introducing image-text matching with hard negative mining also greatly improves the model. This demonstrates the effectiveness of our unified-training framework with Multiway Transformer. In addition, experimental results show that masked language modeling positively contribute to our model. Please refer to the supplementary material for more ablation studies.

| Models | NLVR2 | |
|---|---|---|
| | dev | test-P |
| Local hard negative mining [24] | 77.70 | 77.95 |
| Global hard negative mining (ours) | **79.54** | **79.48** |

Table 6: Global hard negative mining improves the model. We perform experiments using 32 V100 GPUs for the base-size model. The batch size per GPU is 32, and the total batch size is 1024. Local hard negative mining samples hard negatives from training examples of the single GPU (32 examples), while global hard negative mining uses training examples gathered from all GPUs as the candidates (1024 examples).

**Global Hard Negative Mining** Different from ALBEF [24], which samples hard negatives from training examples of the single GPU (named as local hard negative mining). We perform hard negative mining from more candidates by gathering training examples of all GPUs (named as global hard negative mining). As shown in Table 6, our global hard negative mining brings significant improvements.

## 5 Conclusion

In this work, we propose a unified vision-language pretrained model VLMO, which jointly learns a dual encoder and a fusion encoder with a shared Multiway Transformer backbone. Multiway Transformer introduces a pool of modality experts to encode modality-specific information, and aligns different modalities using the shared self-attention module. The unified pre-training with Multiway Transformer enables the model to be used as a dual encoder for efficient vision-language retrieval, or as a fusion encoder to model cross-modal interactions for classification tasks. We also show that stagewise pre-training that leverages large-scale image-only and text-only corpus greatly improves vision-language pre-training. Experimental results demonstrate that VLMO outperforms previous state-of-the-art models on various vision-language classification and retrieval benchmarks.

In the future, we would like to work on improving VLMO from the following perspectives:

- We will scale up the model size used in VLMO pre-training.

- We are also interested in fine-tuning VLMO for vision-language generation tasks, such as image captioning, following the method proposed in UniLM [12].

- We are going to explore to what extent vision-language pre-training can help each other modality, especially as the shared Multiway Transformer backbone naturally blends in text and image representations.

- We can extend the proposed model to integrate more modalities (e.g., speech, video, and structured knowledge), supporting general-purpose multimodal pre-training.

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
