# Supplementary Material: Unified Vision-Language Pre-Training with Mixture-of-Modality-Experts

**Hangbo Bao**[1,*] **Wenhui Wang**[2], **Li Dong**[2], **Qiang Liu**[2], **Owais Khan Mohammed**[2],
**Kriti Aggarwal**[2], **Subhojit Som**[2], **Songhao Piao**[1], **Furu Wei**[2]
[1]Harbin Institute of Technology, [2]Microsoft Corporation
https://aka.ms/msragi

## A    Supplementary Experiments

### A.1    Ablation Study of Multiway Transformer

Table 1 presents the ablation study of shared self-attention module used in Multiway Transformer for encoding image patches and text tokens. We compare shared self-attention with separate self-attention, which encodes image patches and text tokens using different attention parameters on the first $L-F$ layers. The shared self-attention used in Multiway Transformer achieves better performance. The shared self-attention module helps VLMO learn the alignment of different modalities, and fuse images and text at bottom layers for classification tasks.

### A.2    Evaluation on Retrieval Task with Rerank

Following ALBEF [1] and BLIP [2], we conduct experiments on retrieval tasks with rerank. We perform finetuning with image-text contrastive and image-text matching losses. During inference, VLMO is first used as a dual encoder to obtain top-$k$ candidates, then the model is used as a fusion encoder to rerank the candidates. As shown in Table 2, VLMO-Large achieves competitive performance compared with BLIP which uses more data.

## B    Supplementary Hyperparameters

### B.1    Hyperparameters for Pre-Training

The vision-language pre-training of base-size model takes about two days using 64 Nvidia Tesla V100 32GB GPU cards, and the large-size model takes about three days using 128 Nvidia Tesla V100 32GB GPU cards.

For the text-only pre-training data, we use English Wikipedia and BookCorpus [5]. AdamW [3] optimizer with $\beta_1 = 0.9$, $\beta_2 = 0.98$ is used to train the models. The maximum sequence length is set to 196. The batch size is 1024, and the peak learning rate is 2e-4. We set the weight decay to 0.01. For the base-size model, we train the model for 500k steps. The large-size model is trained for 200k steps.

### B.2    Hyperparameters for Vision-Language Classification Fine-Tuning

**Visual Question Answering (VQA)**    We fine-tune the models for 10 epochs with 128 batch size. The peak learning rate is 3e-5 for the base-size model, and 1.5e-5 for the large-size model. Following SimVLM [4], the input image resolution is $480 \times 480$. For VLMO-Large++, we use $768 \times 768$ image resolution. Using $768 \times 768$ resolution brings about 0.3 improvement than $480 \times 480$ resolution.

---

*Contribution during internship at Microsoft.

| Transformer | NLVR2 | | Flickr30k | |
|---|---|---|---|---|
| | dev | test-P | TR | IR |
| Separate Self-Attention | 78.92 | 78.95 | 94.63 | 86.88 |
| **Multiway Transformer (Shared Self-Attention)** | **80.13** | **80.31** | **95.17** | **87.25** |

Table 1: Ablation study of the shared self-attention module used in Multiway Transformer. We experiment with separate attention on the first $L-F$ layers, which encodes image patches and text tokens using different attention parameters.

| Model | # Pretrain Images | Flickr30K (1K test set) | | | | | |
|---|---|---|---|---|---|---|---|
| | | TR | | | IR | | |
| | | R@1 | R@5 | R@10 | R@1 | R@5 | R@10 |
| BLIP [2] | 129M | 97.4 | 99.8 | 99.9 | 87.6 | **97.7** | 99.0 |
| **VLMO-Large** | 4M | **97.7** | **99.9** | **100.0** | **87.8** | 97.7 | **99.1** |

Table 2: Fine-tuning results of text-retrieval (TR) and image-retrieval (IR) with rerank on Flickr30K.

**Natural Language for Visual Reasoning (NLVR2)** For results of Table 1, the models are fine-tuned for 10 epochs with 128 batch size. The peak learning rate of the base-size and large-size models are set to 5e-5 and 3e-5, respectively. The input image resolution is $384 \times 384$. For ablation experiments, we fine-tune the models for 10 epochs with 128 batch size, and choose learning rates from {5e-5, 1e-4}. The input image resolution is $224 \times 224$. All the ablation results of NLVR2 are averaged over 3 runs.

### B.3 Hyperparameters for Vision-Language Retrieval Fine-Tuning

**COCO** We fine-tune the base-size model for 20 epochs and large-size model for 10 epochs with 2048 batch size. The peak learning rate is 2e-5 for the base-size model and 1e-5 for the large-size model. The input image resolution is $384 \times 384$.

**Flickr30K** For results of Table 2, the base-size and large-size models are fine-tuned for 40 epochs with a batch size of 2048 and a peak learning rate of 1e-5. We use the fine-tuned model on COCO as the initialization. The input image resolution is $384 \times 384$. For all ablation experiments, we fine-tune the models for 10 epochs with 1024 batch size. The peak learning rate is set to 5e-5, and the input image resolution is $224 \times 224$.