# OpenReview forum: "VLMo: Unified Vision-Language Pre-Training with Mixture-of-Modality-Experts"
_NeurIPS.cc/2022/Conference — NeurIPS 2022 Accept_

### Official Review · Reviewer_SVes · 2022-07-10

**Rating:** 6
**Confidence:** 5
**Soundness:** 3 good
**Presentation:** 3 good
**Contribution:** 3 good

**Summary:**

This paper proposes a new method called VLMo for multimodal pretraining. VLMo consists of a dual encoder and a fusion encoder with Mixture-of-Experts, where experts of different modalities process information of specified modality or modality combination. This method achieves state-of-the-art performance in several multimodal downstream tasks.

**Questions:**

1. Why does this work uses such a high resolution of 768 * 768 only for VQA, but uses 224*224 or 384 * 384 for the other tasks?
2. Different from the common practice, VLMo adds image-text contrastive learning to the pretraining tasks, which use two towers to process information from different modalities separately, incompatible with the fusion, and I believe that there should be 4 forward operations （2 forwards in the image and text encoder for contrastive learning, 1 for ITM, and 1 for MLM） for a single batch. Does this cause increase in training costs? Is it possible to provide an ablation study on the pretraining tasks?


**Limitations:**

This paper did not point out limitations. It is suggested that the paper explain about lack of analysis in the expert design, lack of capabilities in generation tasks, and lack of further analysis about scaling, especially data scaling.

**Strengths And Weaknesses:**

Strengths:
1. The method achieves state-of-the-art performance in several multimodal downstream tasks.
2. The method creatively incorporates mixture-of-experts with multimodal representation learning, which bring significant improvements in downstream tasks.

Weaknesses:
1. Lack of analysis of the mixture-of-experts. As this is the key innovation of the paper, which is somewhat novel and to the my best of my knowledge the first time to see modality expert in multimodal pretraining, the paper should demonstrate clearly why it is an important design, how it contributes to the performance, how different experts function and cooperate to reach a better effect than the conventional fusion methods.
2. Unclear introduction to the model details: a. The MoE is essentially not the conventional MoE [1] with routing, so there should be clarification to avoid misunderstanding. b. There should be a clear demonstration of the number of parameters. As there are 3 FFN experts in the model, the transformer layers of VLMo-large should consist of at least 700M parameters, let alone the embedding and other parameters. Directly stating that it is a large-size model without clarification misleads readers. c. Based on my understanding, though the model is much larger, it did not bring much increase in computation costs. This should be illustrated clearly. Also, the paper should clarify which layers to use vision and text experts for different modalities, and which layers to process them with the VL experts, instead of using words like "bottom layers" and "top layers".
3. The design of stagewise pretraining requires further explanation as there are many details, e.g., why is such stage design? how about training them jointly? why is the self attention fixed in the second stage? The ablation study should be comprehensive, and make more fair comparisons with the baselines.
4. VLMo achieves stunning performance in VQA, which surpasses even all the current methods as far as I know, but I find from the supplementary details that the VLMO-large++ for VQA uses a high resolution of 768 * 768. This requires explanation or ablation study. From the presented results, readers cannot figure out where the improvements over VLMo-Large come from, scaling data or scaling resolution.
5. This work should implement the model on more downstream tasks, including visual entailment, referring expression comprehension, etc. They are essential benchmarks for multimodal pretrained models. Also, this work should compare VLMo with the state-of-the-art methods (3 months before the NeurIPS submission), including BLIP [2] and OFA [3]. Especially for the cross-modal retrieval, VLMo should be compared with BLIP. Furthermore, as VLMo is pretrained with image-text contrastive learning, it is encouraged to conduct zero-shot cross-modal retrieval experiments.

[1]. Shazeer, N., Mirhoseini, A., Maziarz, K., Davis, A., Le, Q., Hinton, G., & Dean, J. (2017). Outrageously large neural networks: The sparsely-gated mixture-of-experts layer. arXiv preprint arXiv:1701.06538.
[2]. Li, J., Li, D., Xiong, C., & Hoi, S. (2022). Blip: Bootstrapping language-image pre-training for unified vision-language understanding and generation. arXiv preprint arXiv:2201.12086.
[3]. Wang, P., Yang, A., Men, R., Lin, J., Bai, S., Li, Z., ... & Yang, H. (2022). Unifying architectures, tasks, and modalities through a simple sequence-to-sequence learning framework. arXiv preprint arXiv:2202.03052.

---

> ### Author Response · Authors · 2022-08-02
> **Response to Reviewer SVes**
>
> Thanks for your valuable suggestions!
>
> Q1: Why uses 768x768 for VQA, but uses 224x224 or 384x384 for other tasks? \
> A1: For VQA, we follow SimVLM and use 480x480 resolution for VLMo-Base and VLMo-Large to have a fair comparison. We only use 768x768 resolution for VLMo-Large++ since some previous works use a larger resolution. For example, the input image size of VinVL is 800x1333. Using 768x768 resolution brings about 0.3 improvement than 480x480 resolution. \
> For other tasks, we also follow previous works for fair comparisons, such as ViLT and ALBEF, and report results with 384x384 resolution to compare with them. We use 224x224 resolution for all ablation studies to reduce the computation cost.
>
> Q2: There should be 4 forward operations for a single batch. Does this increase training costs? Provide an ablation study on the pretraining tasks? \
> A2: Yes, there are 4 forward operations for a single batch. The training cost increases with more pre-training tasks, but we find that all three pre-training tasks positively contribute to the model. Please refer to Table 4 for the ablation study on the pre-training tasks.
>
> Q3: Lack of analysis of the mixture-of-experts. \
> A3: Thanks for the suggestion! The detailed ablation studies about the performance contributions can be found in Table 4 (i.e., the [4]&[5]&[6] runs in the table).
>
> Q4: Unclear introduction to the model details. \
> A4: We will make it clear about the difference with conventional MoE and add the number of parameters for each model. We also called VLMo-large because the computation cost (e.g., FLOPs) is the same as conventional large-size Transformers. VL experts are used in top two layers of VLMo-Base and top three layers of VLMo-Large as stated in Section 4.1. We will also make it clear in the revised version as you suggest.
>
> Q5: Design of stagewise pretraining. Why is stage design? how about training them jointly? why is the self attention fixed in the second stage? \
> A5: 1) Stage design enables the model to utilize the pretrained weights of vision Transformer to reduce the training cost. We have tried to train them jointly and observe similar performance. 2) Although the self-attention part is frozen, we found that the language model performance remains similar. In contrast, if we do not freeze the self-attention parameters in the second stage, the model will suffer from catastrophic forgetting of vision knowledge learned in the first stage. We observed a performance drop on the downstream vision-language tasks without freezing the self-attention parameters.
>
> Q6: VLMO-large++ uses 768 * 768 resolution for VQA. The improvements over VLMo-Large come from scaling data or scaling resolution. \
> A6: For VQA, we follow SimVLM and use 480x480 resolution for VLMo-Base and VLMo-Large to have a fair comparison. We only use 768x768 resolution for VLMo-Large++ since some previous works use a larger resolution. For example, the input image size of VinVL is 800x1333. Using 768x768 resolution brings about 0.3 improvement than 480x480 resolution. VLMo-Large++ also outperforms previous SOTA with smaller resolution. The improvements are mainly from scaling data and batch size. We will add the results as you suggest.
>
> Q7: This work should test the model on more downstream tasks. This work should compare VLMo with BLIP and OFA. Especially for the cross-modal retrieval with BLIP. It is encouraged to conduct zero-shot cross-modal retrieval experiments. \
> A7: We will compare with BLIP and OFA, and add experiments of visual entailment, referring expression comprehension, zero-shot retrieval in the revised version as you suggest. The table below presents the comparison of VLMo and BLIP on the fine-tuned image-text retrieval (Flickr30k). Since BLIP first selects topk candidates and then rerank them using the fusion module, we also first use VLMo as a dual encoder to select topk candidates and then use the same model as a fusion encoder to rerank the candidates. VLMo-Large trained on 4M images slightly outperforms BLIP trained on 129M images.
> |Model| # Pretrain Images|TR R@1|TR R@5|TR R@10|IR R@1|IR R@5|IR R@10|
> |-|-|-|-|-|-|-|-|
> | BLIP-Large | 129M | 97.4| 99.8| 99.9|87.6|**97.7**|99.0|
> | VLMo-Large | 4M| **97.7** | **99.9** |**100.0** |**87.8** |**97.7** |**99.1** |
>
> Q8: Lack of capabilities in generation tasks. \
> A8: We fine-tune a base-size VLMo on image captioning task by employing masked fine-tuning as in s2s-ft. The table below shows the results on COCO benchmark. VLMo achieves competitive performance. We will add the results to the revised version.
> |Model|BLEU@4|METEOR|CIDEr|SPICE|
> |-|-|-|-|-|
> |OSCAR-Base|36.5|30.3|123.7|23.1|
> |VinVL-Base|38.2|30.3|129.3|23.6|
> |METER-Base|38.8|30.0|128.2| 23.0|
> |SimVLM-Base|39.0|**32.9**|134.8| 24.0|
> |VLMo-Base|**40.3**|31.2|**136.6**| **24.3**|

---

> ### Comment · Reviewer_SVes · 2022-08-09
> **Feedback to rebuttal**
>
> In general, I think you have answered my questions quite well. Q1 and Q2 answer my concerns about increased costs, and make me better understand the contribution of the setups. As to other issues like better introduction and more experiments, I am convinced that you will improve the quality of the paper. What still bothers me are still the setup of stages and also how experts contribute. I understand these are not simple problems to tackle, and I hope that in your future work you can dive into these fundamental issues. I decide to raise my score.

---

### Official Review · Reviewer_oGKY · 2022-07-10

**Rating:** 6
**Confidence:** 4
**Soundness:** 3 good
**Presentation:** 3 good
**Contribution:** 4 excellent

**Summary:**

This paper focuses on Vision-Language Pre-training (VLP), in which they propose a unified vision-language pretrained model VLMO that jointly learns a dual encoder (i.e., the text and image encoders) and a fusion encoder (i.e., the multimodal encoder).

Specifically, the authors introduce the Mixture-of-Modality-Experts (MOME) Transformer to encode different modalities,  where each block contains a pool of modality-specific experts to  capture modality-specific information and a shared self-attention layer to learn the alignments between different modalities. Moreover, it can utilize large amounts of image-only and text-only data to perform stagewise pre-training. The pre-trained model can be used as a fusion encoder for classification tasks, or fine-tuned as a dual encoder for retrieval tasks.



**Questions:**

N/A

**Strengths And Weaknesses:**

Strengths:

1. The motivation of the proposed VLMO model makes sense to me, which can alleviate the limitations of existing single-stream and two-stream VLP models.

2. The paper is generally well organized, and mostly easy to follow. Although the pre-training tasks are generally not new, the proposed Mixture-of-Modality-Experts (MOME) Transformer is relatively novel.

3. Experimental results demonstrate the effectiveness of the proposed pre-training framework and stagewise pre-training strategies.


Weaknesses:

I have several concerns as follows:

1.  In the pre-training stage, the proposed model is first trained with a MRM with the image-only data, followed by a MLM with the text-only data. The motivation is not clear. Is it possible to perform MLM first followed by MRM? Or is it better to perform alternative training?

2. The downstream tasks only include image-text classification/retrieval tasks, but fail to consider many other representative VL generation tasks such as image captioning.

---

> ### Author Response · Authors · 2022-08-02
> **Response to Reviewer oGKY**
>
> Thanks for your valuable suggestions!
>
> Q1: In the pre-training stage, the proposed model is first trained with a MRM with the image-only data, followed by a MLM with the text-only data. Is it possible to perform MLM first followed by MRM? Or is it better to perform alternative training? \
> A1: We have tried to perform alternative training on image-only and text-only data, and observe similar performance with stage-wise training. Compared with alternative training, stage-wise training effectively leverages the pretrained vision models to reduce the computation cost. ViLT has shown that initializing from a pretrained vision model (ViT) achieves better performance than a pretrained language model (BERT). Inspired by ViLT, we perform vision pre-training first. We will also try MLM training first and add these ablation studies in the revised version.
>
> Q2: The downstream tasks only include image-text classification/retrieval tasks, but fail to consider many other representative VL generation tasks such as image captioning. \
> A2: Thanks for the suggestion! We fine-tune a base-size VLMo on image captioning task by employing masked fine-tuning as in s2s-ft [1]. The table below shows the results of different base-size models on COCO benchmark. VLMo also achieves competitive performance. We will add the results to the revised version.
> | Model       | BLEU@4 | METEOR | CIDEr | SPICE  |
> |-------------|--------|--------|-------|--------|
> | OSCAR-Base  | 36.5   | 30.3   | 123.7 | 23.1   |
> | VinVL-Base  | 38.2   | 30.3   | 129.3 | 23.6   |
> | METER-Base  | 38.8   | 30.0   | 128.2 | 23.0   |
> | SimVLM-Base | 39.0   | **32.9**   | 134.8 | 24.0   |
> | VLMo-Base   | **40.3**   | 31.2   | **136.6** | **24.3**   |
> [1] s2s-ft: Fine-Tuning Pretrained Transformer Encoders for Sequence-to-Sequence Learning

---

### Official Review · Reviewer_Z9fk · 2022-07-11

**Rating:** 7
**Confidence:** 2
**Soundness:** 3 good
**Presentation:** 3 good
**Contribution:** 3 good

**Summary:**

The paper introduces an architecture which supports multiple modalities (text, image and text-image), via  a novel transformer module (MoME), in which blocks consist of modality specific experts with shared self-attention.
The method achieves new SOTA on vision language classification tasks (VQA and NLVR2), as well as text retrieval and image retrieval, on MS-COCO 5k and Flickr 30K.
The model can be pre-trained stage-wise, using text-only and image-only large scale databases.


**Questions:**

How does this model compare to  the Flamingo model of Alayrac et al?
How would the model perform on Meta-Dataset of Triantafillou et al? Would the retrieval benefit from text pre-training? Would image pre-training improve results, due to the different data distributions and increased number of classes?
Are the Vision / Language and Vision-Language experts trained independently? Would it benefit the subsequent tasks if these experts were pre-trained upfront?

**Strengths And Weaknesses:**

Strengths:
* support for text-only, image-only and text-image pretraining, which allows leveraging existing datasets, reducing the need  for large datasets with both modalities labeled.
* image-text contrastive learning used for pre-training;
* selection via multi-headed self attention of which expert to use for subsequent processing.
* detailed ablation study.

Weaknesses:
* 4M to 1B images for pre-training leads to ~6% improvement. It would be interesting to understand if similar accuracy could be achieved with less data, or, conversely, get a higher improvement from 250x data.
* (minor) No details about the architecture of the experts (it seems these are feed forward networks).

---

> ### Author Response · Authors · 2022-08-02
> **Response to Reviewer Z9fk**
>
> Thanks for your valuable suggestions!
>
> Q1: It would be interesting to understand if similar accuracy could be achieved with less data, or, conversely, get a higher improvement from 250x data. \
> A1: Thanks for the suggestion! We will try to perform ablation studies on image-text pairs of different sizes in the future.
>
> Q2: Details about the architecture of the experts (it seems these are feed forward networks). \
> A2: Yes. The experts are feed-forward networks as in standard Transformer, which consists of two linear transformations with an activation. We will add the details in the revised version.
>
> Q3: How does this model compare to the Flamingo model of Alayrac et al? \
> A3: For the fine-tuning results on the dataset reported by both two models (VQAv2), VLMo-Large++ outperforms Flamingo by 0.68 (82.78 vs. 82.1) with a much smaller model size (565M vs. 80B). Besides fine-tuning, Flamingo also achieves great success in few-shot learning on a wide range of vision-language tasks. VLMo focuses on fine-tuning on downstream tasks in this work, and we will explore the few-shot capability of VLMo in the future.
>
> Q4: How would the model perform on Meta-Dataset of Triantafillou et al? \
> A4: Meta-Dataset is a large-scale few-shot classification benchmark for training and testing few-shot models. VLMo focuses on fine-tuning on downstream tasks in this work. We will explore the few-shot capability of VLMo on Meta-Dataset.
>
> Q5: Would the retrieval benefit from text pre-training? \
> A5: Yes. Please refer to the ablation study in Table 3. Introducing text pre-training improves the model on Flickr30k dataset.
>
> Q6: Would image pre-training improve results, due to the different data distributions and increased number of classes? \
> A6: Yes. We have compared image pre-training on ImageNet1k and ImageNet21k. We find that using ImageNet21k achieves better performance on downstream vision-language tasks given its larger size and more classes.
>
> Q7: Are the Vision / Language and Vision-Language experts trained independently? \
> A7: We first train vision expert in image-only pre-training, language expert in text-only pre-training. Then three experts are jointly optimized in the vision-language pre-training stage.
>
> Q8: Would it benefit the subsequent tasks if these experts were pre-trained upfront? \
> A8: Yes. Pre-training these experts brings significant improvement. For example, we observe more than 32 point improvement on downstream NLVR2 task (83.3 vs. 51.1).

---

### Meta-Review · Area_Chair_Vdy8 · 2022-08-24

**Recommendation:** Accept
**Confidence:** Certain

**Metareview:**

After the rebuttal and discussion, this paper unanimously receives positive rates. As the reviews show satisfaction on the authors’ feedback, the final draft needs to respect it accordingly. Additionally, reviewers suggest several post-rebuttal points, so the final draft may need to clarify them, including the comparison with Flamingo, the setup of stages and contributions of experts.

**Award:**

No

---

### Decision · Program_Chairs · 2022-09-14

Accept